# Using Point of Care Testing to estimate influenza vaccine effectiveness in the English primary care sentinel surveillance network

Simon de Lusignan[1]*, Uy Hoang[1], Harshana Liyanage[1], Manasa Tripathy[1], Julian Sherlock[1], Mark Joy[1], Filipa Ferreira[1], Javier Diez-Domingo[2], Tristan Clark[3]

1 Nuffield Department of Primary Care Health Sciences, University of Oxford, Oxford, United Kingdom,
2 Vaccine Research Department, FISABIO-Public Health, Valencia, Spain, 3 Academic Unit of Clinical and Experimental Sciences, University of Southampton, Southampton, United Kingdom

* simon.delusignan@phc.ox.ac.uk

## Abstract

### Introduction

Rapid Point of Care Testing (POCT) for influenza could be used to provide information on influenza vaccine effectiveness (IVE) as well as influencing clinical decision-making in primary care.

### Methods

We undertook a test negative case control study to estimate the overall and age-specific (6 months-17 years, 18–64 years, ≥65 years old) IVE against medically attended POCT-confirmed influenza. The study took place over the winter of 2019–2020 and was nested within twelve general practices that are part of the Oxford-Royal College of General Practitioners (RCGP) Research and Surveillance Centre (RSC), the English sentinel surveillance network.

### Results

648 POCT were conducted. 193 (29.7%) of those who were swabbed had received the seasonal influenza vaccine. The crude unadjusted overall IVE was 46.1% (95% CI: 13.9–66.3). After adjusting for confounders the overall IVE was 26.0% (95% CI: 0–65.5). In total 211 patients were prescribed an antimicrobial after swab testing. Given a positive influenza POCT result, the odds ratio (OR) of receiving an antiviral was 21.1 (95%CI: 2.4–182.2, p = <0.01) and the OR of being prescribed an antibiotic was 0.6 (95%CI: 0.4–0.9, p = <0.01).

### Discussion

Using influenza POCT in a primary care sentinel surveillance network to estimate IVE is feasible and provides comparable results to published IVE estimates. A further advantage is that near patient testing of influenza is associated with improvements in appropriate antiviral and antibiotic use. Larger, randomised studies are needed in primary care to see if these trends are still present and to explore their impact on outcomes.

**Data Availability Statement:** Data cannot be shared publicly because of it is owned by the Oxford-RCGP RSC and its participating practices. Data are available from the University of Oxford

Institutional Data Access Committee (contact via simon.delusginan@phc.ox.ac.uk) for researchers who meet the criteria for access to confidential data. The data underlying the results presented in the study are available from https://orchid.phc.ox.ac.uk/. Further enquires about the RCGP RSC network and data requests can be found on the following website https://orchid.phc.ox.ac.uk/index.php/orchid-data/ or by emailing the following address practiceenquiries@phc.ox.ac.uk.

**Funding:** The work described here is part of Development of Robust and Innovative Vaccine Effectiveness (DRIVE) an EU funded project, part of the Innovative Medicines Initiative (IMI) project Grant agreement number: 777363 https://www.imi.europa.eu/ The funders play no role in the study design, data collection and analysis, decision to publish, or preparation of this manuscript.

**Competing interests:** SdeL receives research funding via the University of Surrey from Eli Lilly Co., GlaxoSmithKline, Takeda, AstraZeneca and Novo Nordisk Ltd. TC has also taken part in advisory board meetings for Roche and Janssen, and is a member of independent datamonitoring committees for trials sponsored by Roche. This does not alter our adherence to PLOS ONE policies on sharing data and materials. All other authors have declared no competing interests

## Introduction

Influenza is a major cause of clinical and public health burden [1]. It is estimated to account for 11.5% of all episodes of respiratory infection in the UK, with over 50 000 patients requiring hospital admission and at least 2000 deaths per year [2]. Complications are more common in older and younger age groups, over 65 years and under one year respectively [3].

Vaccination is considered as the most effective means for preventing influenza and its complications [4]. A population vaccination coverage target of at least 75% in the elderly population and among risk groups is recommended by the World Health Organization (WHO) [5].

The vaccine requires reformulating annually to match with the characteristics of the circulating influenza viruses which undergo frequent genetic and antigenic changes [6]. Thus influenza vaccine effectiveness (IVE) is assessed annually and observed IVE varies year-to-year [5, 7].

IVE is interpreted as the proportionate reduction in disease among the vaccinated group in real world conditions as opposed to efficacy in ideal conditions, such as a clinical trial [8]. IVE thus varies depending on whether it is measured in secondary or primary care settings [9].

Regulatory agencies such as the European Medicines Agency (EMA) now require vaccine manufacturers to undertake studies in different settings [10]. This includes studies in real world settings such as primary care to provide specific IVE data including product (brand) specific IVE studies as part of their post-licensure requirements, rather than only relying on annual clinical immunogenicity trials of vaccine.

In the last few years rapid molecular point of care test (POCT) platforms for influenza have become widely available in primary care [11]. These highly accurate tests use nucleic acid amplification tests such as reverse transcription polymerase chain reaction (RT-PCR) which had previously been reserved for use in centralised laboratories [12]. This has allowed rapid, accurate pathological confirmation of influenza infection in the community, and is crucial to undertaking IVE studies in primary care.

We have previously shown testing for influenza using rapid molecular POCT machines is feasible in primary care and associated with improvements in appropriate antiviral and antibiotic use. Study practices provided POCT machines performed more tests than other virology sampling practices when their practice population size and respiratory virus infection rates were taken into account. POCT machines also influenced clinical prescribing practices. Patients with a positive influenza POCT test were significantly less likely to be prescribed an antibiotic and significantly more likely to be prescribed antiviral medication (odds ratios of 0.4, 95%CI: 0.19–0.78; and 14.1, 95%CI: 2.85–70.0 respectively) compared to those with a negative influenza POCT result [13].

We hypothesised that the widespread use of influenza POCT in primary care may also contribute to real world evidence on IVE. In this study we use influenza POCT to estimate seasonal IVE for the first time in a primary care sentinel surveillance network. Our secondary aim was to estimate the effect of POCT testing on antimicrobial (antibiotic and antiviral) prescribing.

## Materials and methods

A test-negative case control study was undertaken to estimate the overall and age-specific (6 months-17 years, 18–64 years, ≥65 years old) IVE against medically attended POCT-confirmed influenza.

### Study setting and population

The study took place between October 2019 when influenza viruses was first detected in the sentinel surveillance network and finished early in March 2020 as a result of the national

COVID-19 pandemic when the sentinel system moved to a process of remote virology patient self-sampling only [14].

It was nested in the English national influenza surveillance network run by the Oxford-Royal College of General Practitioners (RCGP) Research and Surveillance Centre (RSC) [15]. This is one of the longest established primary care sentinel networks in Europe [16]. The RCGP RSC has been collecting the data used in this study for national surveillance for some years, including providing feedback via dashboards to improve data quality [17]. Previous work has shown that the age and sex distribution of patients in the sentinel network is broadly similar to the English national census distribution [15].

Twelve general practices, from RCGP RSC, with a registered population of 184,813 patients were recruited to the study. Practices were provided with a leaflet explaining the study to staff and eligible patients; and were encouraged to display a poster about the study in their waiting areas. Clinicians in the study practices were encouraged to identify potentially eligible patients aged 6 months and above exhibiting symptoms compatible with influenza-like illness (ILI) and with no contraindication for influenza vaccination for influenza POCT testing.

ILI was defined by the RCGP RSC case definition, which combines key features of the European Centre for Disease Control (ECDC) and WHO definition [18–20].

The Abbott ID Now POCT machine was used for this study. It is a small desktop size POCT machine, that uses an isothermal nucleic acid amplification technology. Its diagnostic accuracy was reported in a systematic review, which reported sensitivity in adults of 80.3% (95% confidence intervals [CI] = 63.7 to 90.8%) and 68.5% (95% CI = 40.2 to 87.2%) for influenza A and B, respectively [11]. Vos et al reported a pooled sensitivity of 81.6% (95% CI = 75.4 to 87.9%) for influenza A and B combined and pooled specificity of 94.0% (95% CI = 86.0 to 100%). This compares with a pooled sensitivity of all rapid molecular test for influenza of 90.9% (95% CI = 88.7 to 93.1%) and pooled specificity of 96.1% (95% CI = 94.2 to 97.9%) [21].

## Ethical statement

Potentially eligible patients over the age of 18 were provided with information about the study by their general practitioner or a study clinician. They were asked for written consent to participate. For potentially eligible children under the age of 18, they were provided with age appropriate information sheet on the study and their parent or legal guardian was asked for written consent for them to participate in the study.

The study was approved by the English National Research Ethics Committees (REC) (Integrated Research Application System ref: 252081; REC ref: 19/WM/0015).

## Data analysis

The primary outcome of interest was POCT virologically-confirmed influenza in the study population.

Patients with POCT virologically-confirmed influenza were defined as cases. Each positive test result was classified by influenza type (A and B) although no information on subtype or lineage was available as part of this study. Controls were deemed as those with ILI who were POCT negative for influenza.

Crude and confounder-adjusted IVE and 95% confidence intervals were estimated using the standard approach based on comparing the odds ratio (OR) of vaccination among influenza-positive study participants with the odds of vaccination among influenza-negative study participants [22].

$$IVE = (1 - OR) \times 100\%.$$

 

Confounder-adjusted IVE estimates were derived from logistic regression models. A complete cases analysis was performed as a sensitivity analysis. The following set of covariates were collected from the routine data extracted from the electronic health record and used for confounder adjustment of IVE with backward stepwise elimination to find the best model fit [22].

- Age

- Sex

- Ethnicity, reported in five categories, white, Asian, black, other, or mixed, and maximised using an ontology [23].

- Socio-economic status, measured using the index of multiple deprivation (IMD) [24]. This is a nationally available measure assigned based on post code.

- Any of the following chronic underlying conditions—chronic pulmonary disease, cardiovascular disease, diabetes, liver disease, renal disease, neurologic/neuromuscular conditions, treatment-induced immunosuppression and disease-induced immunosuppression

- Number of GP visits in the 12 months prior to the study period describing a study subject's healthcare seeking behaviour

- Number of hospitalisations in the 12 months prior to the study period were used as proxy for the severity of the chronic conditions

- Influenza vaccination in previous influenza seasons (at least one)

- Pregnancy

- Use of influenza antivirals

- Pneumococcal vaccination

- Death

Our sample size calculation suggested that we needed at least 100 cases, ideally more to produce an estimate of IVE. This is shown in our protocol [25] and built on our feasibility study of the previous season which did not include calculation of IVE. We calculated the minimal detectable overall VE (1) with 80% power (1 – β) and a two-sided 95% confidence coefficient (1 – α/2) for case-control studies using 'cases to controls' ratio of 1:1, 1:2 and 1:4 with the number of cases varying from 100 to 4000, while assuming overall vaccination coverages of 5%, 15%, 30% and 50%.

## Results

In total 648 swabs were taken of which 128 (19.8%) were positive for influenza A or B on swab testing with the POCT machine. Table 1 shows that 263 (40.6%) swabs were taken from males. 66 (10.2%) were aged ≥65 years. 396 (61.1%) were of white ethnicity and 44 (6.3%) were in the most deprived quintile. 175 (27%) swabs were taken from patients who had any underlying risk factors for influenza.

Approximately three swabs were taken per practice per week over the 19 week duration of the study. The average swabbing rate for POCT practices was 0.4, which compares favourably with our previous influenza POCT study in primary care [13].

### Influenza vaccine effectiveness

193 (29.7%) of those who were swabbed had received the seasonal influenza vaccine. Vaccinated patients were more likely to be older, female, have an underlying risk factor for

**Table 1. Characteristics of the patients who had a positive versus negative influenza POCT result.**

| | All swabs (n = 648) | Influenza +ve (n = 128) | Influenza -ve (n = 520) |
|---|---|---|---|
| **Sex** | | | |
| Males | 263 (40.6%) | 60 (46.9%) | 203 (39.0%) |
| Females | 385 (59.4%) | 68 (53.1%) | 317 (61.0%) |
| **Age** | | | |
| 0–17 years | 233 (36.0%) | 55 (43.0%) | 178 (34.2%) |
| 18–64 years | 349 (53.8%) | 64 (50.0%) | 285 (54.8%) |
| ≥65 years | 66 (10.2%) | 9 (7.0%) | 57 (11.0%) |
| **Ethnicity** | | | |
| White ethnicity | 396 (61.1%) | 76 (59.4%) | 320 (61.5%) |
| All other ethnicity | 66 (10.2%) | 13 (10.2%) | 53 (10.2%) |
| Ethnicity unknown | 186 (28.7%) | 39 (30.4%) | 147 (28.3%) |
| **IMD Quintile** | | | |
| 1 (most deprived) | 41 (6.3%) | 15 (11.7%) | 26 (5.0%) |
| 2 | 64 (9.9%) | 15 (11.7%) | 49 (9.4%) |
| 3 | 120 (18.5%) | 27 (21.1%) | 93 (17.9%) |
| 4 | 188 (29.0%) | 32 (25.0%) | 156 (30.0%) |
| 5 (least deprived) | 216 (33.3%) | 36 (28.1%) | 180 (34.6%) |
| Unknown | 19 (3.0%) | 3 (2.4%) | 16 (3.1%) |
| **Risk group** | | | |
| Y | 175 (27.0%) | 28 (21.9%) | 147 (28.3%) |
| N | 473 (73.0%) | 100 (78.1%) | 373 (71.3%) |
| **Average number of GP visits in last 12 months** | 27.6 | 20.7 | 29.3 |
| | (95% CI: 25.7–29.5) | (95% CI: 17.1–24.3) | (95% CI: 27.1–31.4) |
| **Average number hospitalisations in last 12 months** | 1.4 | 1.3 | 1.4 |
| | (95% CI: 1.3–1.5) | (95% CI: 1.1–1.5) | (95% CI: 1.3–1.5) |
| **Influenza vaccination in last season** | | | |
| Y | 181 (27.9%) | 27 (21.1%) | 154 (29.6%) |
| N | 467 (72.1%) | 101 (78.9%) | 366 (70.4%) |
| **Pregnant** | | | |
| Y | 11 (1.7%) | 1 (1%) | 10 (2.0%) |
| N | 637 (98.3%) | 127 (99%) | 510 (98.0%) |
| **Pneumococcal vaccination** | | | |
| Y | 177 (27.3%) | 31 (24.2%) | 146 (28.1%) |
| N | 471 (72.7%) | 97 (75.8%) | 374 (71.9%) |

influenza, have a greater average number of GP visits in last 12 months and more likely to have had pneumococcal vaccination (see Table 2).

The ratio of the odds of vaccination among influenza-positive study participants compared with the odds of vaccination among influenza-negative study participants was 0.53 (95% CI: 0.34–0.86).

Crude unadjusted overall IVE was 46.1% (95% CI: 13.9–66.3). After adjusting for confounders the overall IVE was 26.0% (95% CI: 0–65.5). Crude unadjusted, age stratified IVE and adjusted, age stratified IVE results are shown in Table 3.

## Antimicrobial prescribing

Six patients in total were prescribed an antiviral medication in the 7 days following their POCT test. Five (83.3%) had had a positive POCT test for influenza. Table 4 shows the characteristics of those prescribed antivirals versus other patients.

**Table 2. Characteristics of the patients who were sampled and those who had received seasonal influenza vaccination.**

| | All swabs (n = 648) | Vaccinated patients (n = 193) | Unvaccinated patients (n = 455) |
|---|---|---|---|
| **Sex** | | | |
| Males | 263 (40.6%) | 66 (34.2%) | 197 (43.3%) |
| Females | 385 (59.4%) | 127 (65.8%) | 258 (56.7%) |
| **Age** | | | |
| 0–17 years | 233 (36.0%) | 61 (31.6%) | 172 (37.8%) |
| 18–64 years | 349 (53.8%) | 77 (39.9%) | 272 (59.8%) |
| ≥65 years | 66 (10.2%) | 55 (28.4%) | 11 (2.4%) |
| **Ethnicity** | | | |
| White ethnicity | 396 (61.1%) | 130 (67.4%) | 266 (58.5%) |
| All other ethnicity | 66 (10.2%) | 20 (10.4%) | 46 (10.1%) |
| Ethnicity unknown | 186 (28.7%) | 43 (22.2%) | 143 (31.4%) |
| **IMD Quintile** | | | |
| 1 (most deprived) | 41 (6.3%) | 9 (4.7%) | 32 (7.0%) |
| 2 | 64 (9.9%) | 20 (10.4%) | 44 (9.7%) |
| 3 | 120 (18.5%) | 35 (18.1%) | 85 (18.7%) |
| 4 | 188 (29.0%) | 56 (29.0%) | 132 (29.0%) |
| 5 (least deprived) | 216 (33.3%) | 71 (36.8%) | 145 (31.9%) |
| Unknown | 19 (3.0%) | 2 (1.0%) | 17 (3.7%) |
| **Risk group** | | | |
| Y | 175 (27.0%) | 116 (60.1%) | 59 (13.0%) |
| N | 473 (73.0%) | 77 (39.9%) | 396 (87.0%) |
| **Average number of GP visits in last 12 months** | 27.6 | 41.7 | 21.6 |
| | (95% CI: 25.7–29.5) | (95% CI: 37.3–46.0) | (95% CI: 19.9–23.3) |
| **Average number hospitalisations in last 12 months** | 1.4 | 1.6 | 1.4 |
| | (95% CI: 1.3–1.5) | (95% CI: 1.3–1.8) | (95% CI: 1.2–1.5) |
| **Influenza vaccination in last season** | | | |
| Y | 181 (27.9%) | 135 (69.9%) | 46 (10.1%) |
| N | 467 (72.1%) | 58 (30.1%) | 409 (89.9%) |
| **Pregnant** | | | |
| Y | 11 (1.7%) | 7 (3.6%) | 4 (1.0%) |
| N | 637 (98.3%) | 186 (96.4%) | 451 (99.0%) |
| **Pneumococcal vaccination** | | | |
| Y | 177 (27.3%) | 73 (37.8%) | 104 (22.9%) |
| N | 471 (72.7%) | 120 (62.2%) | 351 (77.1%) |

**Table 3. Age stratified influenza vaccine effectiveness.**

| Crude age stratified IVE | | |
|---|---|---|
| **Age groups** | **IVE** | **95% CI** |
| **0–17 years** | 44.7 | (0–74.1) |
| **18–64 years** | 46.1 | (13.9–66.3) |
| **≥65 years** | NA | NA |
| Adjusted, age stratified IVE | | |
| **Age group** | **IVE** | **95% CI** |
| **0–17 years** | 63.5 | (0–88.7) |
| **18–64 years** | 24.9 | (0–87.2) |
| **≥65 years** | NA | NA |

**Table 4. Characteristics of the patients who were sampled and those who had received antivirals.**

| | All swabs (n = 648) | Received antivirals (n = 6) | Not received antivirals (n = 642) |
|---|---|---|---|
| **Sex** | | | |
| Males | 263 (40.6%) | 3 (50%) | 260 (40.5%) |
| Females | 385 (59.4%) | 3 (50%) | 382 (59.5%) |
| **Age** | | | |
| 0–17 years | 233 (36.0%) | 1 (16.7%) | 232 (36.1%) |
| 18–64 years | 349 (53.8%) | 5 (83.3%) | 344 (53.6%) |
| ≥65 years | 66 (10.2%) | 0 | 66 (10.3%) |
| **Ethnicity** | | | |
| White ethnicity | 396 (61.1%) | 3 (50%) | 393 (61.2%) |
| All other ethnicity | 66 (10.2%) | 1 (16.7%) | 65 (10.1%) |
| Ethnicity unknown | 186 (28.7%) | 2 (33.3%) | 184 (28.7%) |
| **IMD Quintile** | | | |
| 1 (most deprived) | 41 (6.3%) | 0 | 41 (6.4%) |
| 2 | 64 (9.9%) | 0 | 64 (9.9%) |
| 3 | 120 (18.5%) | 4 (66.6%) | 116 (18.1%) |
| 4 | 188 (29.0%) | 1 (16.7%) | 187 (29.1%) |
| 5 (least deprived) | 216 (33.3%) | 1 (16.7%) | 215 (33.5%) |
| Unknown | 19 (3.0%) | 0 | 19 (3.0%) |
| **Risk group** | | | |
| Y | 175 (27.0%) | 3 (50%) | 172 (26.8%) |
| N | 473 (73.0%) | 3 (50%) | 470 (73.2%) |
| **Average number of GP visits in last 12 months** | 27.6 | 28.8 | 27.5 |
| | (95% CI: 25.7–29.5) | (95% CI: 19.5–38.2) | (95% CI: 25.6–29.5) |
| **Average number hospitalisations in last 12 months** | 1.4 | 1.3 | 1.4 |
| | (95% CI: 1.3–1.5) | (95% CI: 0.7–2.0) | (95% CI: 1.3–1.5) |
| **Influenza vaccination in last season** | | | |
| Y | 181 (27.9%) | 3 (50.0%) | 178 (27.7%) |
| N | 467 (72.1%) | 3 (50.0%) | 464 (72.3%) |
| **Pregnant** | | | |
| Y | 11 (1.7%) | 0 | 11 (2.0%) |
| N | 637 (98.3%) | 6 (100%) | 631 (98.0%) |
| **Pneumococcal vaccination** | | | |
| Y | 177 (27.3%) | 3 (50.0%) | 174 (27.1%) |
| N | 471 (72.7%) | 3 (50.0%) | 468 (72.9%) |

205 patients in total were prescribed an antibiotic medication in the 7 days following their POCT test. 30 (14.6%) had had a positive POCT test for influenza. An antibiotic was more likely to be prescribed for older patients with underlying risk factors for influenza and those who had a history of more visits to the GP in the last 12 months, see Table 5.

50.7% (n = 175/345) of patients received antibiotics following a negative influenza POCT test. The odds of being prescribed an antibiotic given a positive result was 0.6 (95% CI: 0.4–0.9) compared with a negative test.

In comparison, 4% (5/123) of patients received an antiviral following a positive influenza test. The odds of receiving an antiviral given a positive result was 21.1 (95% CI: 2.4–182.2) compared with a negative test.

**Table 5. Characteristics of the patients who were sampled and those who had received antibiotics.**

| | All swabs (n = 648) | Received antibiotics (n = 205) | Not received antibiotics (n = 443) |
|---|---|---|---|
| **Sex** | | | |
| Males | 263 (40.6%) | 82 (40%) | 181 (40.9%) |
| Females | 385 (59.4%) | 123 (60%) | 262 (59.1%) |
| **Age** | | | |
| 0–17 years | 233 (36.0%) | 54 (26.3%) | 179 (40.4%) |
| 18–64 years | 349 (53.8%) | 115 (56.1%) | 234 (52.8%) |
| ≥65 years | 66 (10.2%) | 36 (17.6%) | 30 (6.8%) |
| **Ethnicity** | | | |
| White ethnicity | 396 (61.1%) | 21 (10.2%) | 45 (10.2%) |
| All other ethnicity | 66 (10.2%) | 127 (62.0%) | 269 (60.7%) |
| Ethnicity unknown | 186 (28.7%) | 57 (27.8%) | 129 (29.1%) |
| **IMD Quintile** | | | |
| 1 (most deprived) | 41 (6.3%) | 14 (6.8%) | 27 (6.1%) |
| 2 | 64 (9.9%) | 22 (10.7%) | 42 (9.5%) |
| 3 | 120 (18.5%) | 37 (18.0%) | 83 (18.7%) |
| 4 | 188 (29.0%) | 60 (29.3%) | 128 (28.9%) |
| 5 (least deprived) | 216 (33.3%) | 68 (33.2%) | 148 (33.4%) |
| Unknown | 19 (3.0%) | 4 (2.0%) | 15 (3.4%) |
| **Risk group** | | | |
| Y | 175 (27.0%) | 78 (38.0%) | 97 (21.9%) |
| N | 473 (73.0%) | 127 (62.0%) | 346 (78.1%) |
| **Average number of GP visits in last 12 months** | 27.6 | 33.5 | 24.82 |
| | (95% CI: 25.7–29.5) | (95% CI: 29.7–37.2) | (95% CI: 22.7–26.9) |
| **Average number hospitalisations in last 12 months** | 1.4 | 1.5 | 1.4 |
| | (95% CI: 1.3–1.5) | (95% CI: 1.3–1.7) | (95% CI: 1.2–1.5) |
| **Influenza vaccination in last season** | | | |
| Y | 181 (27.9%) | 76 (37.1%) | 105 (23.7%) |
| N | 467 (72.1%) | 129 (62.9%) | 338 (76.3%) |
| **Pregnant** | | | |
| Y | 11 (1.7%) | 4 (2.0%) | 7 (1.6%) |
| N | 637 (98.3%) | 201 (98.0%) | 436 (98.4%) |
| **Pneumococcal vaccination** | | | |
| Y | 177 (27.3%) | 53 (25.9%) | 124 (28.0%) |
| N | 471 (72.7%) | 152 (74.1%) | 319 (72.0%) |

## Discussion

The is the first time, to our knowledge that IVE has been estimated in a primary care sentinel surveillance using network using POCT machines. We have also confirmed that testing for influenza using POCT machines is associated with improvements in appropriate antimicrobial use.

### Strengths and weakness

The non-randomised design is a limitation of our study, along with its short duration, and small sample size. This was a result of the study having to cease just after halfway through the 2019/2020 influenza season following the COVID-19 pandemic. This resulted in the wide confidence intervals around our IVE and OR point estimates and has restricted our ability to

make conclusions from stratified analysis including by influenza virus type and patient demographic characteristics. Additionally, information about influenza subtype was not available from the POCT machines used in this study and restricted interpretation of our results, especially those pertaining to the effectiveness of the seasonal influenza vaccine. Lastly, information about the duration of respiratory illness was not collected before swab testing, thus this limits the conclusions regarding the appropriate use of antiviral medications following POCT.

A strength of the study was that it was conducted amongst practices which were nested in the RCGP RSC English sentinel surveillance network and which had been also been involved in POCT sampling in the previous year. This ensured that practice staff had experience with using the machines, minimising the number of spoiled samples. It also allowed a comparison of the performance of practices using POCT for influenza testing with other practices in the sentinel network that participate in the usual virology sampling programme conducted by Public Health England (PHE).

## Comparison with existing literature

The figures calculated from this study are comparable to the results of published IVE estimates from primary care, including IVE analysis from studies conducted by the ECDC and studies conducted as part of the DRIVE European initiative [26, 27]. This compares with our previous study which found that the odds ratio of receiving an antiviral was 14.1 (95% CI = 2.9 to 70.0, P<0.001) and of receiving an antibiotic was 0.4 (95% CI = 0.2 to 0.8, P = 0.01) given a positive influenza POCT test [13]. Our study reinforces previous findings about the impact of the results of rapid near patient testing for influenza on prescribing practices in non-randomised studies including our own recent study [13, 28].

Given the wide confidence interval of our crude (46.1% (95% CI:13.9–66.3)) and adjusted (26.0% (95% CI: 0–65.5)) IVE, it unsurprisingly fits with other estimates for the same season. IECDC presented estimates from six European studies, covering 10 countries and calculated an IVE between 29% to 61% for all ages in the primary care setting in 2019/20 [26]. The Centre for Disease Control (CDC) in the US calculated the overall IVE of 45% (95% CI = 36%–53%) in the outpatient setting in 2019/20 [29].

For comparison, the DRIVE (Development of Robust and Innovative Vaccine Effectiveness) study undertook a pooled analysis in 2019/20 of data from four primary care sites in Europe using test negative designs that examined 2372 subjects including patients from the UK of which 77 were vaccinated cases. They found the IVE against any influenza in children less than 17y was 64% (95%CI 44–80) for any vaccine [27]. This is comparable with the results from this study where the adjusted IVE against any influenza in children less than 17y was 63.5% (95%CI 0–88.7).

## Implications for research

This study provides the first evidence, to our knowledge, for the use of POCT to estimate IVE within a primary care sentinel network. Further studies however are required with larger sample sizes to ensure that robust point estimates can be obtained for overall and age-specific IVE. Studies that would allow brand/ manufacturer specific IVE estimates to be calculated would also be preferable.

## Author Contributions

**Conceptualization:** Simon de Lusignan, Uy Hoang, Tristan Clark.

**Data curation:** Uy Hoang, Harshana Liyanage, Julian Sherlock.

**Formal analysis:** Uy Hoang.

**Funding acquisition:** Simon de Lusignan.

**Investigation:** Uy Hoang, Harshana Liyanage, Julian Sherlock.

**Methodology:** Simon de Lusignan, Uy Hoang, Mark Joy.

**Project administration:** Simon de Lusignan, Uy Hoang, Harshana Liyanage, Manasa Tripathy, Filipa Ferreira, Tristan Clark.

**Resources:** Simon de Lusignan, Javier Diez-Domingo.

**Software:** Julian Sherlock.

**Supervision:** Simon de Lusignan, Filipa Ferreira.

**Validation:** Julian Sherlock.

**Writing – original draft:** Uy Hoang.

**Writing – review & editing:** Simon de Lusignan, Uy Hoang, Harshana Liyanage, Manasa Tripathy, Mark Joy, Filipa Ferreira, Javier Diez-Domingo, Tristan Clark.

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
