## [Decision Letter · Decision Letter 0]

28 Jan 2021

PONE-D-20-39506

Using Point of Care Testing to estimate influenza vaccine effectiveness in the English primary care sentinel surveillance network

PLOS ONE

Dear Dr. de Lusignan,

Thank you for submitting your manuscript to PLOS ONE. After careful consideration, we feel that it has merit but does not fully meet PLOS ONE’s publication criteria as it currently stands. Therefore, we invite you to submit a revised version of the manuscript that addresses the points raised during the review process.

Please look at the reviewers suggestions to revise and prepare the manuscript to be accepted. They are few but necessary to be addressed.

We look forward to receiving your revised manuscript.

Kind regards,

Ricardo Q. Gurgel, PhD

Academic Editor

PLOS ONE

Journal Requirements:

3. Our staff editors have determined that your manuscript is likely within the scope of our Call for Papers on Influenza. This editorial initiative is headed by PLOS ONE Guest Editors Dr. Meagan Deming and Dr. Deshayne Fell. The Collection encompasses research on influenza prevention on every level, including in vitro, translational, behavioral, and clinical studies; disease and immunity modelling; as well as new approaches to influenza prevention. Additional information can be found on our announcement page: https://collections.plos.org/call-for-papers/influenza/.

Currently, your manuscript is included in the group of papers being considered for this call. Please note that being considered for the Collection does not require additional peer review beyond the journal’s standard process and will not delay the publication of your manuscript if it is accepted by PLOS ONE. We would greatly appreciate your confirmation that you would like your manuscript to be considered for this Collection by indicating this in your next cover letter. If you would prefer to remove your manuscript from collection consideration, please specify this in your cover letter.

"SdeL receives research funding via the University of Surrey from Eli Lilly Co., GlaxoSmithKline, Takeda, AstraZeneca and Novo Nordisk Ltd. TC has also taken part in advisory board meetings for Roche and Janssen, and is a member of independent data-monitoring committees for trials sponsored by Roche. All other authors have declared no competing interests."

7. Thank you for submitting the above manuscript to PLOS ONE. During our internal evaluation of the manuscript, we found significant text overlap between your submission and the following previously published works, some of which you are an author:

- https://www.drive-eu.org/wp-content/uploads/2018/12/DRIVE_D7.1_Core-protocol-for-test-negative-design-studies_1.1.pdf

-https://eprints.soton.ac.uk/443229/1/BJGP2020.pdf

- https://www.pharmaceutical-journal.com/news-and-analysis/research-briefing/point-of-care-testing-for-influenza-could-improve-antimicrobial-use-feasibility-study-concludes/20208259.article?firstPass=false

Please revise the manuscript to rephrase the duplicated text, cite your sources, and provide details as to how the current manuscript advances on previous work. Please note that further consideration is dependent on the submission of a manuscript that addresses these concerns about the overlap in text with published work.

Reviewers' comments:

Reviewer's Responses to Questions

**Comments to the Author**

1. Is the manuscript technically sound, and do the data support the conclusions?

Reviewer #1: Yes

Reviewer #2: Yes

2. Has the statistical analysis been performed appropriately and rigorously? 

Reviewer #1: Yes

Reviewer #2: No

3. Have the authors made all data underlying the findings in their manuscript fully available?

Reviewer #1: Yes

Reviewer #2: No

4. Is the manuscript presented in an intelligible fashion and written in standard English?

Reviewer #1: Yes

Reviewer #2: Yes

5. Review Comments to the Author

Reviewer #1: This trial is a fascinating real-life evaluation. The point of care test is an interesting way to evaluate patients and make possible more rapid therapeutics. Only 4% (5/123) of the patients .received antiviral therapeutic. It would be interesting to determine why so few antiviral therapeutics.

There were not any evaluation of disease severity among the groups vaccinated and non-vaccinated with a positive test.

Other data that would be interested in being included would be the incidence of Influenza A or B.

Could the vaccine be more useful to one of those Influenza types? A or B?

Was the vaccine used trivalent influenza Vaccine or quadrivalent?

Reviewer #2: Thank you for the opportunity to review this interesting manuscript. In this study the authors used influenza Point-of-care test to estimate seasonal of vaccine effectiveness in a primary health care surveillance network and the effect of POCT on antimicrobial prescribing.

Overall the study was well-designed and the paper is well written. However, despite efforts, less concern can be raised about the description of the data analysis. More details on how the logistic regression model is missing. For example, what method of logistic regression was used (Forward LR or Backward LR)?

Other point is the need to discuss the findings of vaccine effectiveness and antimicrobial prescription by age group.

In addition, how do the authors discuss the differences in IVE results by age group studied?

6. PLOS authors have the option to publish the peer review history of their article (what does this mean?). If published, this will include your full peer review and any attached files.

Reviewer #1: **Yes: **eitan berezin

Reviewer #2: No

---

## [Author Response · Author response to Decision Letter 0]

11 Feb 2021

From: em.pone.0.713e8a.ec89bae5@editorialmanager.com <em.pone.0.713e8a.ec89bae5@editorialmanager.com> on behalf of PLOS ONE <em@editorialmanager.com>

Sent: Wednesday, February 10, 2021 11:28

To: Simon de Lusignan <simon.delusignan@phc.ox.ac.uk>

Subject: PLOS ONE: Your submission PONE-D-20-39506R1 - [EMID:8cda8bf0fbe23568] 

IMPORTANT: PLEASE DO NOT REPLY TO THIS EMAIL

If you are unable to complete any points that are requested in this email, please explain why in the "Enter Comments" tab of the online submission form prior to re-submitting your manuscript. This will enable us to promptly assess your response and progress your manuscript to an Academic Editor at the earliest opportunity.

PONE-D-20-39506R1

Using Point of Care Testing to estimate influenza vaccine effectiveness in the English primary care sentinel surveillance network

Prof Simon de Lusignan

Dear Prof de Lusignan,

Thank you for submitting your manuscript entitled "Using Point of Care Testing to estimate influenza vaccine effectiveness in the English primary care sentinel surveillance network" to PLOS ONE. Your manuscript files have been checked in-house but before we can proceed we need you to address the following issues:

1. Thank you for providing additional details regarding the restrictions on data sharing. In line with our goal of ensuring long-term data availability to all interested researchers, PLOS’ Data Policy states that authors cannot be the sole named individuals responsible for ensuring data access (http://journals.plos.org/plosone/s/data-availability#loc-acceptable-data-sharing-methods).

Before we proceed with your manuscript, please also provide contact information for a data access committee (i.e. the University of Oxford Institutional Data Access Committee) to which data requests may be sent. This contact should not be an author.

Oxford reply – We have included in our Data Availability Statement a link below to the website where data requests can be submitted to the University of Oxford Institutional Data Access Committee

https://orchid.phc.ox.ac.uk/index.php/orchid-data/

Further enquires about the RCGP RSC network and data requests can also be emailed to the following address

practiceenquiries@phc.ox.ac.uk

2. We note that the Introduction, Results, and Discussion sections of your manuscript still contain significant text overlap with the following previously published works:

- https://www.drive-eu.org/wp-content/uploads/2018/12/DRIVE_D7.1_Core-protocol-for-test-negative-design-studies_1.1.pdf

-https://eprints.soton.ac.uk/443229/1/BJGP2020.pdf

- https://www.pharmaceutical-journal.com/news-and-analysis/research-briefing/point-of-care-testing-for-influenza-could-improve-antimicrobial-use-feasibility-study-concludes/20208259.article?firstPass=false

Oxford reply – The introduction, results and discussion have been substantially revised to remove overlapping text from previous work

Please note that this text overlap must be addressed in order for us to consider your submission further. Please revise the manuscript to eliminate the duplicated text, and - if relevant - provide details as to how the current manuscript advances on previous work.

Your manuscript has been returned to your account. Please log on to PLOS Editorial Manager at https://www.editorialmanager.com/pone/ to access your manuscript.

Your manuscript can be found in the "Revisions Sent Back to the Author" link under the New Submissions menu. After you have made the changes requested above, please be sure to view and approve the revised PDF after rebuilding the PDF to complete the resubmission process. 

Please note that these changes have been requested to comply with submission guidelines and your manuscript will *not* be sent to review until you have fully adhered to our requests. Once your paper has been seen by an Editor we may return it to you for further information or amendments.

We ask that you address this request within 21 days. If you require additional time, please email the journal office. We are happy to grant extensions of up to one month past this due date. If we have not heard from you within 21 days, your manuscript will be withdrawn from Editorial Manager. 

Kind regards,

Kirstin Darroch

PLOS ONE

In compliance with data protection regulations, you may request that we remove your personal registration details at any time. (Use the following URL: https://www.editorialmanager.com/pone/login.asp?a=r). Please contact the publication office if you have any questions.

---

## [Editor Report · Decision Letter 1]

22 Feb 2021

Using Point of Care Testing to estimate influenza vaccine effectiveness in the English primary care sentinel surveillance network

PONE-D-20-39506R1

Dear Dr. de Lusignan,

We’re pleased to inform you that your manuscript has been judged scientifically suitable for publication and will be formally accepted for publication once it meets all outstanding technical requirements.

Kind regards,

Ricardo Q. Gurgel, PhD

Academic Editor

PLOS ONE
---

## [Editor Report · Acceptance letter]

24 Feb 2021

PONE-D-20-39506R1 

Using Point of Care Testing to estimate influenza vaccine effectiveness in the English primary care sentinel surveillance network 

Dear Dr. de Lusignan:

I'm pleased to inform you that your manuscript has been deemed suitable for publication in PLOS ONE. Congratulations! Your manuscript is now with our production department. 

Kind regards, 

on behalf of

Professor Ricardo Q. Gurgel 

Academic Editor

PLOS ONE